# RBM38 Reverses Sorafenib Resistance in Hepatocellular Carcinoma Cells by Combining and Promoting lncRNA-GAS5

**DOI:** 10.3390/cancers15112897

**Published:** 2023-05-24

**Authors:** Xing Gao, Cheng Lu, Ziyu Liu, Yan Lin, Julu Huang, Lu Lu, Shuanghang Li, Xi Huang, Minchao Tang, Shilin Huang, Ziqin He, Xiaomin She, Rong Liang, Jiazhou Ye

**Affiliations:** 1Department of Medical Oncology, Guangxi Medical University Cancer Hospital, Nanning 530021, China; gaoxing@stu.gxmu.edu.cn (X.G.); liuziyu@stu.gxmu.edu.cn (Z.L.); linyan@gxmu.edu.cn (Y.L.); lishuanghang@stu.gxmu.edu.cn (S.L.); huangxi@stu.gxmu.edu.cn (X.H.); huangshilin@stu.gxmu.edu.cn (S.H.); heziqin@stu.gxmu.edu.cn (Z.H.); shexiaomin@stu.gxmu.edu.cn (X.S.); 2Department of Hepatobiliary Surgery, Guangxi Medical University Cancer Hospital, Nanning 530021, China; lucheng@stu.gxmu.edu.cn (C.L.); huangjulu@stu.gxmu.edu.cn (J.H.); tangminchao@stu.gxmu.edu.cn (M.T.)

**Keywords:** drug resistance, hepatocellular carcinoma, lncRNA GAS5, RBM38, sorafenib

## Abstract

**Simple Summary:**

Hepatocellular carcinoma (HCC) represents the fourth leading cause of cancer-related death in the world. Sorafenib, a first-line chemotherapeutic drug that inhibits cell growth and angiogenesis, has become the standard therapy for inoperable advanced HCC. However, while sorafenib improves survival and is cost-effective in advanced HCC patients, its efficacy is severely restricted by the development of drug resistance. As a result, more research is needed to investigate the underlying molecular process of sorafenib resistance and to identify novel molecular targets. This study aimed to determine the mechanisms underlying sorafenib resistance. Herein, we found activation of RBM38 significantly reverses the resistance of HCC cells to sorafenib by restoring the expression of the lncRNA GAS5, indicating that the RBM38–GAS5 signaling pathway is involved in sorafenib resistance. Therefore, RBM38 and the GAS5 may serve as promising therapeutic targets for enhancing the responsiveness of patients with advanced HCC to sorafenib.

**Abstract:**

Background: Hepatocellular carcinoma (HCC) is a life-threatening human malignancy and the fourth leading cause of cancer-related deaths worldwide. Patients with HCC are often diagnosed at an advanced stage with a poor prognosis. Sorafenib is a multikinase inhibitor used as the first-line treatment for patients with advanced HCC. However, acquired resistance to sorafenib in HCC leads to tumor aggression and limits the drug’s survival benefits; the underlying molecular mechanisms for this resistance remain unclear. Methods: This study aimed to examine the role of the tumor suppressor RBM38 in HCC, and its potential to reverse sorafenib resistance. In addition, the molecular mechanisms underlying the binding of RBM38 and the lncRNA GAS5 were examined. The potential involvement of RBM38 in sorafenib resistance was examined using both in vitro and in vivo models. Functional assays were performed to assess whether RBM38: binds to and promotes the stability of the lncRNA GAS5; reverses the resistance of HCC to sorafenib in vitro; and suppresses the tumorigenicity of sorafenib-resistant HCC cells in vivo. Results: RBM38 expression was lower in HCC cells. The IC_50_ value of sorafenib was significantly lower in cells with RBM38 overexpression than in control cells. RBM38 overexpression improved sorafenib sensitivity in ectopic transplanted tumors and suppressed the growth rate of tumor cells. RBM38 could bind to and stabilize GAS5 in sorafenib-resistant HCC cells. In addition, functional assays revealed that RBM38 reversed sorafenib resistance both in vivo and in vitro in a GAS5-dependent manner. Conclusions: RBM38 is a novel therapeutic target that can reverse sorafenib resistance in HCC by combining and promoting the lncRNA GAS5.

## 1. Introduction

Hepatocellular carcinoma (HCC) is the sixth most common malignancy and the fourth leading cause of cancer-related deaths worldwide [1]. In 2018, a total of 841,000 new cases of HCC were reported worldwide, most of which were reported in Eastern Asia [1], with >50% morbidity and mortality rates reported in China [2]. Because a majority of patients with HCC are diagnosed at an advanced stage, such regional therapies as liver resection, liver transplantation, radiofrequency ablation (RFA), transcatheter arterial chemoembolization (TACE), and radiotherapy are ineffective in these patients [3]. Multitargeted tyrosine kinase inhibitors (TKIs), such as sorafenib, can suppress tumor cell proliferation by inhibiting Raf-1, B-Raf, and kinase activity in the Ras/Raf/MEK/ERK signaling pathway. Additionally, sorafenib can target the platelet-derived growth factor receptor beta (PDGFR-β), the vascular endothelial growth factor receptor (VEGFR) 2, and the hepatocyte factor receptor (c-Met), among other proteins, to inhibit tumor angiogenesis [4]. Sorafenib was the earliest approved first-line therapy for advanced HCC, being approved in 2007 [5]; however, limited therapeutic efficacy has been observed in clinical practice since then [6]. In patients with advanced HCC, sorafenib only increased overall survival (OS) by around 3 months when compared to placebo, according to the results of the SHARPE [7] and Oriental [5] studies. HCC is characterized by high heterogeneity and dysregulation of a diverse and complex group of intracellular signaling pathways, factors which impair the response of HCC to sorafenib. As a result, drug resistance is frequently acquired within 6 months [8]. Despite the availability of other TKIs and immune checkpoint inhibitors (ICIs) as alternative therapies [9], sorafenib remains the most important and commonly used first-line treatment for advanced HCC [9,10]. Therefore, exploring the molecular mechanisms underlying sorafenib resistance may help to identify novel therapeutic strategies for improving its response in HCC.

RNA-binding proteins (RBPs) are a group of conserved proteins in mammalian cells, which contain one or more RNA-binding domains (RBDs) [11]. RBPs function by binding to their target RNA and forming ribonucleoprotein (RNP) complexes [12] to control several essential cellular processes, such as RNA splicing, modification, transport, localization, stability, translation and degradation [13,14]. Therefore, RBPs are involved in the post-transcriptional upregulation and downregulation of multiple tumor-suppressor genes and oncogenes, which may lead to tumorigenesis and the progression of human cancers [15]. The RNA-binding motif protein 38 (RBM38) is a member of the RNA recognition motif family of RBPs; the gene which encodes RBM38 is located on chromosome 20q13, and is downregulated in various cancer tissues [16]. RBM38 contains a classical RNA recognition motif (RRM) domain, and binds to transcripts which contain AU- or U-rich elements, which facilitates post-transcriptional regulation of numerous downstream proteins [16]. Therefore, it plays a key role in various biological functions, including cell proliferation, apoptosis, migration, and invasion [17,18]. Jiang et al. [19] reported that compared with wild-type mice, RBM38-deficient mice had a shorter lifespan and died mostly from spontaneous tumors, indicating that RBM38 functions as an intergenic suppressor of tumorigenesis. Recent studies have shown that RBM38 is extensively involved in the occurrence and metastasis of tumors, including in breast [20], renal [21], gastric [22], and liver cancers [23]. RBM38 is potentially maintained by p53 under normal conditions and, in turn, regulates p53 levels under stress conditions. Therefore, the translational regulation of p53 by RBM38 represents a feedback loop for the p53 pathway [24]. Xu et al. [25] reported that RBM38 can independently suppress MDM2 by destabilizing its transcripts after binding to multiple sites in the 3′-untranslated region (3′-UTR) of MDM2, irrespective of the presence of p53. Our previous study revealed that RBM38 plays a key regulatory role in rescuing the functions of the p53–MDM2 axis, and that it suppresses HCC aggressiveness by stabilizing the p53–MDM2 loop function [26]. RBPs are widely involved in anticancer drug resistance. The human antigen R (HuR) can reverse epirubicin-triggered drug resistance in colorectal cancer [27], whereas RBM8A can promote oxaliplatin resistance, and RBMX can promote sorafenib resistance in HCC [28,29]. However, whether RBM38 is involved in anticancer drug resistance remains to be elucidated.

This study aimed to determine the mechanisms underlying sorafenib resistance. To the best of our knowledge, this study is the first to validate the therapeutic potential of RBM38 to reverse sorafenib resistance in HCC through in vitro and in vivo studies. This study offers a novel therapeutic strategy for improving the therapeutic efficacy of sorafenib in HCC.

## 2. Materials and Methods

### 2.1. Cell Lines and Cultures

Human HCC cell lines (HepG2, Hep3B, MHCC97H, 7721, and Huh7) and normal liver cells (L02) were obtained from the Shanghai Institute of Biological Sciences, Chinese Academy of Sciences (Shanghai, China). All cell lines were cultured in Dulbecco’s modified Eagle’s medium (DMEM, Invitrogen, Waltham, MA, USA), supplemented with 10% fetal bovine serum (FBS, Gibco, Thermo Fisher Scientific, Waltham, MA, USA) and 1% penicillin–streptomycin (Gibco), at 37 °C and under 5% CO_2_. The medium was changed every 3–4 days, and the cells were passaged with 0.25% trypsin (Hyclone, Logan, UT, USA).

### 2.2. Establishment of Sorafenib-Resistant HCC Cell Lines

Our previous study revealed that the human HCC cell lines 7721 and HepG2 have relatively low RBM38 expression [26]. To establish sorafenib-resistant cell lines, 7721 and HepG2 cells were treated with lethal concentration 50 (LC_50_) of sorafenib, and the concentration was gradually increased by 10% until the maximum tolerated dose (10 μM) was reached. Resistant cell lines were maintained in the continuous presence of 10 μM sorafenib, which was supplemented every 72 h. Sorafenib resistance was evaluated using the cell counting kit-8 (CCK-8) assay.

### 2.3. Establishment of Stable Cell Lines with RBM38 Overexpression and lncRNA GAS5 Knockdown

Lentiviruses overexpressing the RBM38 sequence (OE), lncRNA GAS5 knockdown (KD), and the negative control (NC) (Vector) were synthesized by Hanyin Biotechnology (Shanghai, China). To establish stable cell lines, 7721 and HepG2 cells were seeded in 6-well plates and transfected with a lentivirus using polybrene the following day. The stable cell lines obtained included the following: 7721-NC, 7721-RBM38-OE, 7721/sora-NC, 7721/sora-RBM38-OE, HepG2-NC, HepG2-RBM38-OE, HepG2/sora-NC, and HepG2/sora-RBM38-OE cells. The efficiency of RBM38 overexpression was verified via Western blotting. Additionally, GAS5 was knocked down in 7721/sora-RBM38-OE and HepG2/sora-RBM38-OE cells, as indicated.

### 2.4. Quantitative Real-Time PCR

Total RNA was extracted from cultured cells using the TRIzol reagent (A-79061; Takara, Dalian, China) according to the manufacturer’s instructions. The extracted mRNA was quantified with a quantitative real-time polymerase chain reaction (RT-qPCR) using SYBR Green PCR Master Mix (Roche, Basel, Switzerland) on an ABI 7500 Fast Real-Time PCR system. The mRNA expression levels were evaluated using the comparative threshold cycle (Ct) method, normalized to those of β-actin, converted to fold changes (2^−ΔΔCt^), and expressed as the n-fold difference relative to the control. The qPCR raw data are shown in Appendix A. 

### 2.5. Western Blotting

The primary antibodies used for Western blotting included anti-RBM38 (ab200403; Abcam, Cambridge, UK), anti-p-GP (cy5669; Abway, Shanghai, China), anti-MRP1 (ba0567; Boster, Wuhan, China) and anti-ABCG (227286-1-AP; Proteintech, Wuhan, China). After three washes, the membranes were incubated with horseradish peroxidase-conjugated secondary antibodies, and protein bands were visualized via electrochemiluminescence (ECL). Band intensities were quantified via densitometric analysis, using β-actin (SANTA CRUZ, Santa Cruz, CA, USA) as the loading control.

### 2.6. Cell Counting Kit-8 Assay

The proliferation rates and viability of the suitably-treated cells were measured using the Cell Counting Kit-8 (CCK-8; Dojindo, Kumamoto, Japan) according to the manufacture’s protocol. The cells were seeded in a 96-well plate at a density of 2000 cells per well, and cultured for 24, 48, 72, 96, and 120 h. At each time point, 100 μL fresh medium and 10 μL CCK-8 solution were added into each well, and the cells were incubated for 3 h. The absorbance at 450 nm was measured using a microplate reader (5082 Grodig, Tecan, Grödig, Austria).

### 2.7. Transwell Assay

The migration and invasion abilities of HCC cells were evaluated using transwell chambers (8 µm pore; CORNING, Corning, NY, USA) coated either with (for the invasion assay) or without (for the migration assay) Matrigel, in accordance with the manufacturer’s protocol. Next, 1 × 10^5^ cells/well were rapidly seeded in the upper chambers in serum-free media, and the lower chambers were filled with complete medium (with 10% FBS). After 24 h incubation, the cells that had invaded into the lower chambers were fixed with 4% paraformaldehyde for 15 min, and then stained with 0.1% crystal violet solution at room temperature for 30 min. The stained cells were washed with PBS and counted under an inverted microscope (Olympus Corporation; magnification, 100×) using the ImageJ software version 1.8 (National Institute of Health). The number of migratory or invasive cells was counted in 3 random fields (200×), and the average was calculated.

### 2.8. Cell Apoptosis Assay

Cells were seeded in 6-well plates at a density of 1 × 10^6^ cells/well and cultured for 24 h. Apoptotic cells were collected and stained using an apoptosis detection kit based on phycoerythrin-conjugated annexin V (FXP018-100; 4A Biotech, Beijing, China) according to the manufacturer’s instructions. Completely stained cells were analyzed via flow cytometry (FACSCalibur; BD Biosciences, Franklin Lakes, NJ, USA).

### 2.9. Cell Cycle Analysis

The cultured cells were harvested and washed twice with cold PBS, then fixed overnight with 70% ethanol at 4 °C. The following day, after incubating with 200 μL RNase A (Sigma, St. Louis, MO, USA) and 500 μL PI staining buffer in the dark at room temperature for 30 min, the cells were acquired using a FACSCalibur flow cytometer (FACSCalibur; BD Biosciences), and data were analyzed using the FlowJo software (version 10.0).

### 2.10. RNA Immunoprecipitation

An RNA immunoprecipitation (RIP) assay was performed using the Magna RIP kit (Millipore, Billerica, MA, USA) in accordance with to the manufacturer’s instructions. Cells were then promptly lysed using the RIPA lysis buffer, and all cell lysates were incubated using an RIP buffer containing magnetic beads which had been conjugated with both the Ago2 antibody and the normal anti-rabbit IgG. Co-precipitated RNAs were obtained and analyzed via RT-qPCR.

### 2.11. RNA Pulldown Assay 

RNA pulldown assay was performed using RNA Pulldown Assay Kit (Thermo Fisher Scientific, Waltham, MA, USA). Biotinylated GAS5 probe was conjugated to streptavidin magnetic beads, followed by the incubation with protein lysates for 1 h. Beads were washed briefly three times and boiled in SDS buffer, and the retrieved protein was determined by western blot analys.

### 2.12. Dual-Luciferase Reporter Assay

Full-length GAS5 and fragmented GAS5 (160–473 bp and 387–698 bp) sequences were incorporated into pmirGLO basic plasmids, which were transfected into 293T cells using a Lipofectamine 2000 kit. After 48 h of transfection, the luciferase activity of the cells was measured using a dual-luciferase reporter gene assay kit (Promega, Madison, WI, USA).

### 2.13. Actinomycin D Assay

7721/sora-NC and 7721/sora-RBM38-OE cells were seeded in 24-well plates and incubated for 24 h. Thereafter, 5 μg/mL actinomycin D (Abcam) was added to the cells to inhibit transcription, and the expression of GAS5 was detected at different time points (0, 4, 8, 12, and 24 h) via RT-qPCR.

### 2.14. Molecular Docking

Data on RBM38 were extracted from the Protein Data Bank (PDB, https://www.rcsb.org/pages/contactus (accessed on 7 May 2022) in a PBD file format [30]. The RNA sequence of lncRNA GAS5 was downloaded from The National Center for Biotechnology Information (NCBI, https://www.ncbi.nlm.nih.gov/ (accessed on 7 May 2022), and its secondary structure was predicted using the RNAfold web server (http://rna.tbi.univie.ac.at/cgi-bin/RNAWebSuite/RNAfold.cgi (accessed on 7 May 2022). RNAComposer was used to predict the three-dimensional structure of GAS5 based on its secondary structure [31]. Hex (version 8.0.0) [32] was used to perform molecular docking, and Pymol (version 2.5) [33] was used to visualize the results.

### 2.15. Establishment of Xenograft-Bearing Nude Mouse Models

This study was approved by the Ethics Committee of Guangxi Medical University and was performed in accordance with the Animal Research: Reporting of In Vivo Experiments (ARRIVE) guidelines. Animal breeding was supervised by a specialist in the animal center. Nude BALB/C mice (age, 5–6 weeks; weight, 18–22 g) were randomly divided into four groups (eight mice in each group). Either stable 7721/sora-RBM38-OE, 7721/sora-NC or 7721/sora-RBM38-OE + GAS5-KD cells (1 × 10^6^ cells in 0.1 mL of PBS) were subcutaneously injected into nude mice, and tumor growth was monitored for 6 weeks. The length and width of tumors were measured using a vernier caliper, and tumor volume was calculated using the following formula: V = (L × W^2^)/2.

### 2.16. Immunohistochemical Staining

Tumor tissues harvested from the mice were fixed with 10% formalin, embedded in paraffin, and cut into thin sections of 3–4 μm. The tissue sections were deparaffinized, rehydrated, and subjected to antigen retrieval. Thereafter, they were blocked with 3% H_2_O_2_ for 10 min and incubated with anti-RBM38, anti-p-GP, anti-MRP1 and anti-ABCG2 antibodies overnight at 4 °C. The following day, the tissue sections were incubated with biotin-conjugated goat anti-rabbit IgG secondary antibody at 25 °C, stained with diaminobenzidine (DAB), counterstained with hematoxylin, and washed with water. 

### 2.17. Statistical Analysis

The SPSS Statistics (version 19.0, IBM Corp., Armonk, NY, USA) and GraphPad Prism (version 8, GraphPad Software, La Jolla, CA, USA) software were used for the statistical analysis of all experimental data. All experiments in this study were performed in triplicate, unless otherwise specified. Both Student’s *t*-test and ANOVA were used to estimate significant differences between groups. A *p*-value of <0.05 was considered statistically significant.

## 3. Results

### 3.1. RBM38 Expression Decreased Sorafenib Resistance and RBM38 Upregulation Promoted Sorafenib Sensitivity in HCC Cells

RBM38 expression was examined in five HCC cell lines and one normal human liver epithelial cell line (Lo2) via RT-qPCR and Western blot. The results revealed that RBM38 mRNA and protein expression was lower in HCC cells than in normal liver cells (*p* < 0.01) (Figure 1A,B). Likewise, as shown in the box plot and violin plot in Appendix A, RBM38 was reduced in HCC samples (case group) relative to the HCC peritumor tissue (control group) in the GSE76427 dataset. To examine the potential role of RBM38 in sorafenib resistance, sorafenib-resistant HCC cell lines (HepG2/sora and 7721/sora) were established by treating cells with sorafenib at different concentrations (Figure 1C). RT-qPCR and Western blotting revealed that RBM38 expression was significantly lower in both HepG2/sora and 7721/sora cells than it was in their corresponding control cells (Figure 1D,E). Consistent with this, as shown in the box plot and violin plot in Appendix A, RBM38 was reduced in HCC sorafenib sensitive samples (case group) relative to the HCC sorafenib resistance samples (control group) in the GSE109211 dataset. These results suggest that lower RBM38 expression is related with sorafenib resistance in HCC cells.

To assess whether RBM38 affects the therapeutic efficacy of sorafenib in HCC cells, HepG2/sora and 7721/sora cells were transfected with RBM38-overexpressing plasmids to establish resistant cell lines overexpressing RBM38 (Figure 1F,G). RBM38 overexpression in 7721/sora and HepG2/sora cells significantly decreased the IC_50_ values (Figure 1H). In addition, Western blotting revealed that the levels of the drug resistance-related proteins ABCG2, MRP1, and P-GP were lower in cells overexpressing RBM38 than in control cells (Figure 1I). Altogether, these results suggest that upregulation of RBM38 promotes sorafenib sensitivity in HCC cells (Figure 1).

### 3.2. RBM38 Reverses Sorafenib Resistance in HCC by Inducing Cell Apoptosis and Suppressing Tumorigenicity in Xenograft Mouse Models

To examine the role of RBM38 in reversing sorafenib resistance, phenotypic experiments related to drug resistance were performed. As shown in Figure 2A, the proliferation rate of cells overexpressing RBM38 (HepG2/sora-RBM38-OE [*p* < 0.01] and 7721/sora-RBM38-OE [*p* < 0.01] cells) was significantly lower than that of the corresponding control cells. Cellular apoptosis was examined using flow cytometry, which revealed a significantly higher proportion of apoptotic and necrotic cells in RBM38-overexpressing cells than in the control cells (Figure 2B). Furthermore, compared with the corresponding control cells, the number of respective HepG2/sora-RBM38-OE and 7721/sora-RBM38-OE cells were higher in the G1 phase (*p* < 0.01) and lower in the S phase (*p* < 0.01) (Figure 2C). In addition, a transwell assay revealed that the migration (*p* < 0.01) and invasion (*p* < 0.01) abilities of HepG2/sora-RBM38-OE and 7721/sora-RBM38-OE cells were significantly lower than those of the corresponding control cells (Figure 2D,E). These results indicate that RBM38 inhibits the proliferation, migration, and invasion capabilities of sorafenib-resistant HCC cells, in addition to promoting apoptosis in vitro.

To examine the role of RBM38 in reversing sorafenib resistance in HCC in vivo, xenograft-bearing nude mouse models were established using 7721/sora-NC and 7721/sora-RBM38-OE cells (Figure 2F). Tumor size was smaller (*p* < 0.05), tumor weight was lower (*p* < 0.01), and tumor growth was slower (Figure 2G,H) in mice treated with 7721/sora-RBM38-OE cells than in those treated with 7721/sora-NC cells. In addition, immunohistochemical (IHC) analysis revealed that the levels of drug resistance-related proteins were significantly lower in tumor tissues of mice treated with 7721/sora-RBM38-OE cells than in those of mice treated with 7721/sora-NC cells (Figure 2I). Altogether, these results indicate that RBM38 can reverse sorafenib resistance in HCC cells, both in vivo and in vitro.

### 3.3. RBM38 Promotes the Stability of lncRNA GAS5 in Sorafenib-Resistant HCC Cells

RBM38 is a member of the RBP family, and its main function is post-transcriptional regulation. As an RBP, RBM38 can bind to several RNAs. Studies have reported that the lncRNA GAS5 is dysregulated in sorafenib-resistant cells, something which is itself closely associated with the poor prognosis of HCC. However, the mechanism of action of GAS5 in regulating sorafenib resistance has remained unclear. Therefore, we selected GAS5 for further analysis, and examined the role of RBM38 in reversing sorafenib resistance.

Molecular docking was performed to analyze the binding between RBM38 and GAS5. The docking energy of RBM38 and GAS5 was −547.96 kJ/mol, which verifies their capacity for targeted binding (Figure 3A). Furthermore, RIP assay revealed that the enrichment of GAS5 in samples obtained via immunoprecipitation with anti-RBM38 antibody was comparable to that of samples obtained via immunoprecipitation with isotype-matched IgG controls (Figure 3B). In addition, protein samples pulled down using the probe obtained after transcription using the GAS5 sense strand contained the RBM38 protein, and protein samples pulled down using the probe obtained after transcription using the GAS5 antisense strand did not contain the RBM38 protein (Figure 3C). These results indicated successful binding between RBM38 and GAS5 in 7721/sora-RBM38-OE cells. Dual-luciferase reporter assay was performed to verify the binding between RBM38 and GAS5. Co-transfection of plasmids with RBM38 overexpression and either full-length or fragmented (387–698 bp) GAS5 significantly increased the luciferase activity of sorafenib-resistant HCC cells (*p* < 0.01); however, co-transfection of plasmids with RBM38 overexpression and fragmented GAS5 (160–473 bp) did not influence the luciferase activity (*p* > 0.05) (Figure 3D). In addition, RT-qPCR revealed that GAS5 expression was lower in HCC cells (HepG2, Hep3B, MHCC97H, and 7721 cells) than in normal liver cells (Lo2 cells, *p* < 0.01), with the exception of Huh7 cells. (Figure 3E). Moreover, RBM38 overexpression in sorafenib-resistant HCC cells enhanced the mRNA expression of GAS5 (Figure 3F). To investigate whether RBM38 stabilizes GAS5, RBM38-overexpressing cells were treated with actinomycin D (an RNA synthesis inhibitor), and RT-qPCR was used to analyze the mRNA expression of GAS5. The results revealed that RBM38 increased the stability of GAS5 (Figure 3G). Altogether, these results suggest that RBM38 promotes the stability of GAS5 in sorafenib-resistant HCC cells.

### 3.4. RBM38 Enhances Sorafenib-Induced Apoptosis in Sorafenib-Resistant HCC Cells in a GAS5-Dependent Manner

To investigate the functional role of RBM38 and GAS5 in sorafenib-resistant HCC cells, sh-lncRNA-GAS5 was transfected into sorafenib-resistant HCC cells that were overexpressing RBM38, for use in loss-of-function experiments. The results of the RT-qPCR verified the low expression of GAS5 in HCC cells (Figure 4A). As shown in Figure 4B, the IC_50_ values of sorafenib were drastically increased in HepG2/sora-RBM38-OE and 7721/sora-RBM38-OE cells after sh-lncRNA-GAS5 transfection (*p* < 0.01). In addition, Western blotting revealed that GAS5 knockdown increased the levels of the resistance-related proteins ABCG2, MRP1, and P-GP (Figure 4C).

Cell viability was assessed using the CCK-8 assay, and a proliferation curve was plotted. The cell viability of the GAS5-knockdown group was higher than that of the control group (*p* < 0.01) (Figure 4D). In addition, GAS5 knockdown reduced the apoptotic rate of sorafenib-resistant HCC cells (*p* < 0.01) (Figure 4E). The number of 7721/sora-RBM38-OE + GAS5-KD cells was lower in the G0/G1 phase and higher in the S phase when compared with the number of 7721/sora-RBM38-OE cells (*p* < 0.01), and the number of HepG2/sora-RBM38-OE + GAS5-KD cells was lower in the G0/G1 phase when compared with the number of HepG2/sora-RBM38-OE cells (*p* < 0.01) (Figure 4F). In addition, transwell assay showed that GAS5 knockdown enhanced the migration and invasion abilities of 7721/sora-RBM38-OE and HepG2/sora-RBM38-OE cells (Figure 4G,H).

To further evaluate the ability of RBM38 to bind to GAS5 and reverse sorafenib resistance, the tumorigenicity of 7721/sora-RBM38-OE and 7721/sora-RBM38-OE + GAS5-KD cells was evaluated in nude mice. GAS5-knockdown or control cells were subcutaneously injected into the mammary fat pads of mice. The results showed that tumor growth was faster in mice that had been treated with 7721-sora-RBM38-OE + GAS5-KD cells. In addition, tumor volume and weight were higher in mice that had been treated with 7721/sora-RBM38-OE + GAS5-KD cells than in control mice (Figure 4I–L). Furthermore, IHC analysis showed that the expression of drug resistance-related proteins was higher in tumor tissues of mice that had been treated with 7721/sora-RBM38-OE + GAS5-KD cells than in those treated with 7721/sora-RBM38-OE cells (Figure 4M). These results strongly suggest that RBM38 binds to and promotes the stability of GAS5, in addition to reversing the resistance of HCC to sorafenib in vitro. Additionally, RBM38 suppresses the tumorigenicity of sorafenib-resistant HCC cells in vivo in a GAS5-dependent manner.

## 4. Discussion

Most patients diagnosed with advanced HCC are not considered for regional therapies such as surgical resection, liver transplantation, radiofrequency ablation, TACE, and radiotherapy, owing to poor prognosis [8]. Moreover, HCC is not sufficiently sensitive to conventional cytotoxic chemotherapeutics owing to multidrug resistance, which limits multiple medication options [34]. Despite significant advances in the management and treatment of patients with advanced HCC over the past decades, the complexity of carcinogenesis and genomic heterogeneity limit the therapeutic efficacy, and the prognosis remains poor [35]. Sorafenib is the most widely used first-line systemic drug in the treatment of HCC [36]. However, patients with HCC initially respond to sorafenib but develop drug resistance during the first 6 months of treatment [37], which is considered a major obstacle contributing to the failure of sorafenib treatment in HCC. Sorafenib exerts anti-proliferative effects primarily by inhibiting the RAF/MEK/ERK signaling pathway. However, cancer cells can maintain cell proliferation by activating alternative signaling pathways, such as EGFR/ErbB and PI3K/Akt/mTOR, to resist sorafenib-induced apoptosis [38]. In addition, under hypoxia, the activation of either HIF-1α or HIF-2α induces VEGF and other proangiogenic factors, so as to confer sorafenib resistance to HCC cells [39]. The underlying mechanisms of sorafenib resistance are complicated and multifactorial. Although biotechnological progress has been achieved in the past few decades, the precise molecular mechanism underlying sorafenib resistance remains unclear.

Aberrant expression of RBPs plays an important role in regulating the oncogenic isoforms that affect cell cycle regulation, cell proliferation, migration, invasion, and multidrug resistance in various cancers, thereby participating in the progression of human cancers [40]. Anderson et al. found that carnosol (compounds present in plant species of the Lamiaceae family, such as rosemary and sage) treatment of Familial dysautonomia (FD) patient-derived fibroblasts increases the expression of RBM38 transcripts and proteins [41]. Deregulation (that is, either downregulation or upregulation) of several RBPs have been reported in several in vivo and in vitro studies, demonstrating that it may be responsible for the response of tumor cells to sorafenib [29,42]. To the best of our knowledge, this study is the first to report the involvement of RBM38 in reversing sorafenib resistance in HCC cells in vitro. The in vivo tumorigenesis assay showed that RBM38 overexpression improved sorafenib sensitivity in ectopic transplanted tumors and suppressed their growth. These findings provide an alternative explanation for the development of drug resistance in HCC and indicate that RBM38 can be used as a biomarker for predicting drug resistance in HCC.

Furthermore, we investigated the role of RBM38 in reversing sorafenib resistance. The results of RIP, luciferase reporter, and RNA stability assays verified that RBM38 can bind to the lncRNA GAS5 and increase its stability sufficiently enough to mediate anticancer drug resistance. GAS5 is a non-protein-coding gene originally isolated from a subtraction complementary DNA (cDNA) library, prepared for screening highly expressed genes in growth-arrested cells [43]. Current evidence suggests that downregulated GAS5 promotes tumorigenesis and the development of various malignancies, such as lung [44], bladder [45], breast [46], gastric [47], cervical [48], ovarian [49], and liver [50,51] cancers. In vitro and in vivo studies have demonstrated that decreased GAS5 expression in human HCC cell lines induces cell proliferation and migration, thus indicating the role of GAS5 in tumor suppression and as a therapeutic target in HCC [50,51]. Moreover, GAS5 is closely related to drug resistance in various malignant tumors, which is consistent with the results of this study. Upregulation of GAS5 can dramatically reverse cisplatin resistance in HCC cells by regulating miR-222 [52], whereas its downregulation can increase doxorubicin resistance in HCC cells by inhibiting the miR-21–PTEN signaling pathway [53]. In this study, RBM38 reversed sorafenib resistance both in vivo and in vitro in a GAS5-dependent manner. Furthermore, functional assays revealed that downregulation of GAS5 promoted the proliferation, migration, and invasion capabilities of sorafenib-resistant HCC cells and reduced apoptosis and senescence, partly by promoting cell cycle progression. Importantly, the lack of growth inhibition in resistant HCC in xenograft models treated with sorafenib accompanied with decreased GAS5 expression suggests that GAS5 mediates the response of HCC to sorafenib. Liu et al. [54] reported that GAS5 is involved in sorafenib resistance. GAS5 functions as a competing endogenous RNA to repress miR-21, which regulates its downstream target SOX5, resulting in the sensitivity of renal cell carcinoma to sorafenib.

This study has some limitations. First, although the molecular mechanisms of the RBM38–lncRNA–GAS5 pathway underlying sorafenib resistance in HCC cells were elucidated via in vitro and in vivo experiments, further clinical verification is required in primary samples from patients with sorafenib-resistant HCC. Second, the exact mechanism through which RBM38 and GAS5 participate in the regulation of sorafenib resistance in HCC warrants further investigation. The precise mechanism may be associated with the role of RBM38 and GAS5 in EGFR activation, c-Jun activation, autophagy, protein kinase B activation, formation of a hypoxic environment, dysregulation of apoptosis, cancer stem cell renewal, and epithelial–mesenchymal transition. Therefore, the mechanism of action of RBM38 in regulating GAS5 and their synergistic involvement in regulating sorafenib resistance requires further investigation.

## 5. Conclusions

Downregulation of RBM38 and GAS5 is a key event in the development of sorafenib resistance in HCC cells. Activation of RBM38 significantly reverses the resistance of HCC cells to sorafenib by restoring the expression of the lncRNA GAS5, indicating that the RBM38–GAS5 signaling pathway is involved in sorafenib resistance. Therefore, RBM38 and the GAS5 may serve as promising therapeutic targets for enhancing the responsiveness of patients with advanced HCC to sorafenib.

## Data Availability

The raw data presented in this study can be found in online repositories (https://www.jianguoyun.com/p/DR_IunMQk4vUChjE2cMEIAA (accessed on 29 December 2022)).

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
