# Peer review of "RBM38 Reverses Sorafenib Resistance in Hepatocellular Carcinoma Cells by Combining and Promoting lncRNA-GAS5"

_cancers, 2023, doi:10.3390/cancers15112897_

Round 1
Reviewer 1 Report
In their study entitled "RBM38 reverses sorafenib resistance in hepatocellular carcinoma cells by combining and promoting lncRNA-GAS5", Gao et al. highlighted the role of RBM38 in the acquisition of sorafenib resistance of HCC cells. This study will be of interest of the researchers working in this field as sorafenivb is the major and most potent anti-HCC treatment nowadays although resistance are almost always oberved. Nonetheless, this study suffers from some major lacks that the authors should tackle before considering it for publication.
Major comments:
1) Overall, there is a lack of precision in the description of the figures. It is sometimes hard to understand the figure, all above for non-specialist. The authors should provide more information especially in the legend of the figures.
2) The statistical test used in this study are not appropriate at all. With a triplicate, a non-parametric test such as Mann-Whitney MUST be used. At least a fourth replicate will be necessary to expect a significant difference. Furthermore, are the error bars corresponding to standard deviation or to sem? This should be indicated in the legend of the figure.
3) The authors compared several HCC cell lines to the Lo2 cell line to pretend that RBM38 and GAS5 expression. However, the usage of LO2 cell line as a surrogate for normal liver cells is controversial (PubMed=26116706). It was shown in 2015 to be a derivative of HeLa cells. Thus, the authors should perform a STR-profiling on their cell line to confirm their identity or change their model to claim a RBM38 decreased expression in HCC cell lines. Fuurthermore, to my opinion, using publicly available RNA-seq data comparing normal vs HCC liver or tumor vs peri-tumor would be much better to tackle this question.
4) Figure 1A and 1C: the authors claim that Western blot were quantified but do not show the results. They have to display the result as for the figure 1A, the lower level of RBM38 is not that obvious by eye.
5) Lane 230-231: "These results suggest that RBM38 plays an important role in regulating sorafenib resistance in HCC cells". This is absolutely untrue at this stage. Here, the authors ONLY show that the acquisition of resistance to sorafenib is associated with a decreased expression of RBM38 without necessarily a causal relationship.
6) The authors should show a WB comparing the level of these drug-resistance protein in parental and resistant cell lines to prove the resistance. Besides, the authors should provide references for these proteins.
7) According to these results, one can expect that RBM38 repression in parental cells could lower their sensitivity to sorafenib. Is it the case?
8) Figure 2D: There is a discrepancy between what the authors write and what the figure shows. There are in figure 2D more cells in OE condition compared to NC condition. The authors should change their conclusion according to the figure.
9) The RIP data are not convincing with a very poor enrichment if we can call it an enrichment. Moreover, it has only been done in 7721 cells and not in HepG2 cells. Can the author reproduce the data in HepG2 cells? To confirm the interaction between GAS5 and RBM38 if it exists, the authors should perform a RNA pulldown assay in both models.
10) Lane 310-312: "In addition, qRT-PCR revealed that GAS5 expression was lower in HCC cells (HepG2, Hep3B MHCC97H, 7721 and Huh7 cells) than in normal liver cells (Lo2 cells) (P < 0.01) (Figure 3D)". The decreased expression of GAS5 is not observed in Huh7 cells. In contrast, we observe an increased expression in this model. The authors should make the statement.
11) What happens in non OE cells when GAS5 is repressed? Does the repression of GAS5 totally rescued the effect of RBM38-OE or is it a partial rescue?
12) Are there any drug increasing RBM38 expression? If yes, the authors should discuss it.
Minor comments:
1) For non-specialist, the authors should be more precise about what they consider to be necrotic and apoptotic cells (Figure 2B). Is the apoptotic rare corresponding to the % Annexin V+ cells?
2) For sake of homogeneity, the authros should keep the same colour legend for NC and OE cells in the whole figure. It is perturbating to have NC cells in red in figure 2G while they appear in blue in 2H-I.
3) Lane 295-296: A reference is missing here to support what the authors says about the role of GAS5 in sorafenib resistance.
4) Lane 369-371: "To further evaluate the ability of RBM38 to bind to GAS5 and reverse sorafenib resistance in vivo, the tumorigenicity of 7721/sora-RBM38-OE and 7721/sora-RBM38-OE+GAS5-KD cells was evaluated in nude mice." The authors did not test the ability of RBM38 to bind GAS5 in vivo. The authors should change the sentence.
Author Response
Dear reviewer:
Thanks very much for taking your time to review this manuscript. I really appreciate all your comments and suggestions! Please find my itemized responses in below and my revisions in the re-submitted files.
Point 1: Overall, there is a lack of precision in the description of the figures. It is sometimes hard to understand the figure, all above for non-specialist. The authors should provide more information especially in the legend of the figures.
Response 1: Thanks for the suggestions. We have added more information especially in the legend of the figures.
Point 2: The statistical test used in this study are not appropriate at all. With a triplicate, a non-parametric test such as Mann-Whitney MUST be used. At least a fourth replicate will be necessary to expect a significant difference. Furthermore, are the error bars corresponding to standard deviation or to sem? This should be indicated in the legend of the figure.
Response 2: Thank you for your comments. We have double-checked our data and recalculated it. As we do not have much experimental data, using non-parametric tests may not account for our data very well. Our data were compared between two groups and qualified for t-test calculation, so for differences in continuous variables between two groups, the t-test was used for analysis; while for differences between more than two groups, such as Cell counting kit-8 assay and Cell cycle analysis were used ANOVA, and P<0.05 was considered a statistically significant difference. In addition, according to the information we have found, repeating the experiment three times will reduce the randomness of the experiment, so we have taken a triplicate approach. If you have a different suggestion, please do not hesitate to let us know. We have corrected the problem with the error bars in the graph. Thank you again for correcting our study.
Point 3: The authors compared several HCC cell lines to the Lo2 cell line to pretend that RBM38 and GAS5 expression. However, the usage of LO2 cell line as a surrogate for normal liver cells is controversial (PMID:26116706). It was shown in 2015 to be a derivative of HeLa cells. Thus, the authors should perform a STR-profiling on their cell line to confirm their identity or change their model to claim a RBM38 decreased expression in HCC cell lines. Furthermore, to my opinion, using publicly available RNA-seq data comparing normal vs HCC liver or tumor vs peri-tumor would be much better to tackle this question.
Response 3: Thank you for your comments on our study. As you mentioned, we searched the GEO database for hepatocellular carcinoma cohort data (GSE76427) containing samples from Case (HCC tumour tissue) and Control (HCC peritumour tissue) and analysed the results (Suppkement 1A-B), which were consistent with our findings that RBM38 expression was reduced in HCC tumour tissue.
Point 4: Figure 1A and 1C: the authors claim that Western blot were quantified but do not show the results. They have to display the result as for the figure 1A, the lower level of RBM38 is not that obvious by eye.
Response 4: We have quantified the Western blot result of Figure 1A and 1C,and we also feel great thanks for your point out.
Point 5: Lane 230-231: "These results suggest that RBM38 plays an important role in regulating sorafenib resistance in HCC cells". This is absolutely untrue at this stage. Here, the authors ONLY show that the acquisition of resistance to sorafenib is associated with a decreased expression of RBM38 without necessarily a causal relationship.
Response 5: Thanks! We have adjusted the description of the article. These results suggest that lower RBM38 expression is related with sorafenib resistance in HCC cells.
Point 6: The authors should show a WB comparing the level of these drug-resistance protein in parental and resistant cell lines to prove the resistance. Besides, the authors should provide references for these proteins.
Response 6: Thank you for your suggestion, on the issue of protein selection, by reviewing the data we found that P-GP protein can reduce the intracellular concentration of sorafenib drug (PMID:23299930), MRP-1 is overexpressed in HCC and contributes to sorafenib resistance, (PMID:35203285), and ABCG2 expression regulates HCC resistance to sorafenib (PMID:32554246). All three proteins are involved in and contribute to sorafenib resistance in HCC, which is the reason for our selection of these proteins.
Point 7: According to these results, one can expect that RBM38 repression in parental cells could lower their sensitivity to sorafenib. Is it the case?
Response 7: Yes, our results of our study showed that RBM38 repression in parental cells could lower their sensitivity to sorafenib.
Point 8: Figure 2D: There is a discrepancy between what the authors write and what the figure shows. There are in figure 2D more cells in OE condition compared to NC condition. The authors should change their conclusion according to the figure.
Response 8: Thank you very much for your advice. We have checked our original experimental data in accordance with your suggested instructions. We apologise for the inadvertent misplacement of the HepG2/sora- RBM38-OE and 7721/sora-RBM38-OE results, but we have carefully checked and corrected the image errors. Thank you again for pointing out our error.
Point 9: The RIP data are not convincing with a very poor enrichment if we can call it an enrichment. Moreover, it has only been done in 7721 cells and not in HepG2 cells. Can the author reproduce the data in HepG2 cells? To confirm the interaction between GAS5 and RBM38 if it exists, the authors should perform a RNA pulldown assay in both models.
Response 9: Thank you for your suggestion. We have added RNA-pulldown experiments on GAS5 and RBM38 in 7721 cells, the results of which are presented in Figure 3C, indicating that RBM38 and GAS5 can bind successfully, which we describe and talk about in the article. Due to time issues, we did not set up the RIP assay and RNA-pulldown assay for RBM38 and GAS5 in HepG2 cells. In a follow-up study, we will focus on this issue and continue to explore it in depth.
Point 10: Lane 310-312: "In addition, qRT-PCR revealed that GAS5 expression was lower in HCC cells (HepG2, Hep3B MHCC97H, 7721 and Huh7 cells) than in normal liver cells (Lo2 cells) (P < 0.01) (Figure 3D)". The decreased expression of GAS5 is not observed in Huh7 cells. In contrast, we observe an increased expression in this model. The authors should make the statement.
Response 10: Thanks! We have adjusted the description of the article. In addition, RT-qPCR revealed that GAS5 expression was lower in HCC cells (HepG2, Hep3B, MHCC97H and 7721 cells) than in normal liver cells (Lo2 cells, P < 0.01), with the exception of Huh7 cells. (Figure 3D).
Point 11: What happens in non OE cells when GAS5 is repressed? Does the repression of GAS5 totally rescued the effect of RBM38-OE or is it a partial rescue?
Response 11: According to your question, we found in our previous published study that GAS5 knockdown enhanced the proliferation of HCC cells in vitro and in vivo and increased the resistance of HCC cells to the chemotherapeutic agent doxorubicin (PMID:31705194), which is similar to our findings. Also in our study, GAS5 knockdown was partially rescued by RBM38-OE.
.Point 12: Are there any drug increasing RBM38 expression? If yes, the authors should discuss it.
Response 12: Thank you for your question. In our query of previous studies, we did find drugs that could increase RBM38: Carnosol, a diterpene present in rosemary, increased the expression of RBM38 in mouse models of familial autonomic dysfunction of patient origin (PMID:35708500). Thank you again for your valuable comments, which have helped us a lot in our research, and we will refine the discussion of this section in the Discussion section.
Minor comments:
Comment 1: For non-specialist, the authors should be more precise about what they consider to be necrotic and apoptotic cells (Figure 2B). Is the apoptotic rare corresponding to the % Annexin V+ cells?
Response 1: Based on the information consulted, we obtained that apoptosis is a form of programmed death triggered by multiple microenvironmental perturbations, including (but not limited to) DNA damage, endoplasmic reticulum stress, reactive oxygen species (ROS) overload, replicative stress and other factors that stimulate its production. Cell death: Cell death manifests as macroscopic morphological changes, showing cytoplasmic shrinkage, chromatin condensation (sequestration), nuclear fragmentation (nuclear rupture) and blistering of the plasma membrane, or as extensive cytoplasmic vacuolisation, again ending in phagocytic uptake and subsequent lysosomal degradation (PMID:29362479). Membrane association protein V (Annexin V) is an abundant cytoplasmic protein that initiates the membrane repair process during cell damage in cell injury. Annexin V and PI double staining is the classical method for flow detection of apoptosis, based on the principle that phosphotidylserine (PS) flips from the inner side of the cell membrane to the surface of the cell membrane at the early stage of apoptosis. Annexin V is a Ca2+-dependent phospholipid-binding protein that binds with high affinity to PS. Fluorescein FITC labelling of Annexin V allows for earlier detection of apoptosis by flow cytometry. Propidium iodide (PI) is a DNA-binding dye that does not penetrate the intact cell membrane of normal or early apoptotic cells, but in mid- to late-stage apoptotic cells and dead cells, PI can penetrate the cell membrane and stain the nucleus. The combination of the two can therefore distinguish between living cells, apoptotic cells, and necrotic cells. This is reflected in the scatter plot of fluorescence intensity (Fig2B. Fig4E), where the four regions correspond to different states of cells, namely Q1 for Annexin V-PI+, which is a cell fragment that no longer has a cell membrane or is otherwise necrotic; Q2 for late apoptotic or necrotic cells Annexin V+/PI+; Q3 for normal (living) cells Annexin V-/PI-; and Q4 for early apoptotic cells Annexin V-/PI-. PI-; Q4 is Annexin V+/PI- for early apoptotic cells.
Comment 2: For sake of homogeneity, the authros should keep the same colour legend for NC and OE cells in the whole figure. It is perturbating to have NC cells in red in figure 2G while they appear in blue in 2H-I.
Response 2: Thanks for your nice suggestions. We have done.
Comment 3: Lane 295-296: A reference is missing here to support what the authors says about the role of GAS5 in sorafenib resistance.
Response 3: Thank you for your suggestion. We read the relevant literature and found that GAS5 overexpression can make sorafenib unresponsive to renal clear cell carcinoma (RCC), while knockdown of GAS5 in then treatment in RCC can promote sorafenib resistance (PMID:31705194).
Comment 4: Lane 369-371: "To further evaluate the ability of RBM38 to bind to GAS5 and reverse sorafenib resistance in vivo, the tumorigenicity of 7721/sora-RBM38-OE and 7721/sora-RBM38-OE+GAS5-KD cells was evaluated in nude mice." The authors did not test the ability of RBM38 to bind GAS5 in vivo. The authors should change the sentence.
Response 4: Thanks for your nice suggestions. We have done.
You can also view in the attachment

Reviewer 2 Report
This study investigated the role of the tumour suppressor RBM38 and its potential to reverse sorafenib resistance in HCC. In addition, molecular mechanisms underlying the binding of RBM38 and the lncRNA GAS5 were examined.
This article found that RBM38 expression was lower in HCC cells. The IC50 value of sorafenib was significantly lower in cells with RBM38 overexpression than in control cells. RBM38 overexpression improved sorafenib sensitivity in ectopic transplanted tumours and suppressed the growth rate of tumour cells. RBM38 could bind to and stabilise GAS5 in sorafenib- resistant HCC cells.
This is a well-written study. However, some issues in this study need to be considered.
Comments
1. The author established the stable cell lines with RBM38 overexpression and lncRNA GAS5 knockdown. Do the author have the cell lines with RBM38 knockdown or suppression or/and lncRNA GAS5 overexpression.
2. Sorafenib is a multi-kinase inhibitor targeting RAF, vascular endothelial growth factor (VEGF) receptor (VEGFR) and platelet-derived growth factor receptor (PDGFR). The authors should investigate the relationship of VEGFR inhibition with sorafenib-resistant HCC cell lines and cell lines with RBM38 overexpression or/and lncRNA GAS5 knockdown.
Author Response
Dear reviewer:
Thanks very much for taking your time to review this manuscript. I really appreciate all your comments and suggestions! Please find my itemized responses in below and my revisions in the re-submitted files.
Point 1: The author established the stable cell lines with RBM38 overexpression and lncRNA GAS5 knockdown. Do the author have the cell lines with RBM38 knockdown or suppression or/and lncRNA GAS5 overexpression.
Response 1: Thank you for your question. Because in our previous study we confirmed that RBM38 is lowly expressed in hepatocellular carcinoma and showed that RBM38 inactivation may promote hepatocellular carcinoma progression by disrupting p53-mdm2 loop function (PMID:30176896).In addition, in previous studies by others, GAS5 was similarly overexpressed in hepatocytes and overexpression of GAS5 was shown to inhibit the M2-like polarization of TAM, thereby suppressing the proliferation and invasion of hepatocellular carcinoma cells (PMID:33146930). Therefore, we did not establish a cell line for RBM38 knockdown or inhibition or/and lncRNA GAS5 overexpression in the present study.
Point 2: Sorafenib is a multi-kinase inhibitor targeting RAF, vascular endothelial growth factor (VEGF) receptor (VEGFR) and platelet-derived growth factor receptor (PDGFR). The authors should investigate the relationship of VEGFR inhibition with sorafenib-resistant HCC cell lines and cell lines with RBM38 overexpression or/and lncRNA GAS5 knockdown.
Response 2: We appreciate your suggestion to our study. RBM38 has been shown to be a tumour suppressor, expressed in a variety of tumours, with a role in inhibiting tumour proliferation and invasion (PMID:32642788). However, we have not yet found any relevant studies illustrating the interaction between RBM38 and VEGFR, and our current study did not address this aspect. For this question, we will experimentally validate this at a later stage. I hope you could understand our difficulties, thank you.
You can also view in the attachment

Reviewer 3 Report
In this manuscript, the authors presented a study evaluating the role of RBM38 in sorafenib resistance using hepatocellular carcinoma (HCC) cell lines. In vitro and in vivo studies showed that RBM38 canceled sorafenib resistance in HCC cell lines through regulation of GAS5 expression and binding.
This study showed interesting results. However, as they mentioned in the discussion, the weak point of this study was the lack of clinical data. They did not show any clinical data showing the relationship between RBM38 expression and sorafenib resistance. They only used human HCC cell lines. They should offer these supporting data using clinical samples.
Other points
In Fig 2C, they showed the cell cycle analysis. In these analyses, cell cycle arrest was observed in cells overexpressing RBM38. However, this analysis did not show an increase in apoptotic cells, in contrast with Fig 2B data. Please explain this inconsistency in these experiments.
Author Response
Dear reviewer:
Thanks very much for taking your time to review this manuscript. I really appreciate all your comments and suggestions! Please find my itemized responses in below and my revisions in the re-submitted files.
Point 1:This study showed interesting results. However, as they mentioned in the discussion, the weak point of this study was the lack of clinical data. They did not show any clinical data showing the relationship between RBM38 expression and sorafenib resistance. They only used human HCC cell lines. They should offer these supporting data using clinical samples.
Response 1: Thank you for pointing out the weakness in our study, which was missing clinical data related to RBM38 expression, which is a major weakness in our study. So we found the cohort associated with sorafenib for hepatocellular carcinoma in the published GEO public database(GEO109211) and explored the expression of RBM38 in both, using response to sorafenib treatment as Case and no response to sorafenib treatment as Control, and the results of our analysis showed that the expression of RBM38 was slightly higher in the Control group than in the Case group, in line with our study results The results of our analysis showed that the expression of RBM38 was slightly higher in the Control group than in the Case group (Suppkement 1C-D), in line with our findings.
Other points,
Point 1: In Fig 2C, they showed the cell cycle analysis. In these analyses, cell cycle arrest was observed in cells overexpressing RBM38. However, this analysis did not show an increase in apoptotic cells, in contrast with Fig 2B data. Please explain this inconsistency in these experiments.
Response 1: Thank you for your question, the relevant literature revealed that the cell cycle typically consists of G0/G1 phase, S phase, G2 phase, and M phase. The main task of G1 phase is DNA synthesis, whereas the main task of S phase is DNA replication. The results of figure 2C show an increase in G1 phase cells and a decrease in S phase cells, indicating that the cells are arrested in G1 phase with a relative decrease in DNA synthesis and replication. Combined with the results of proliferation and apoptosis experiments, it suggests that overexpression of RBM38 will inhibit cell proliferation and induce cell apoptosis.
You can also view in the attachment.

Round 2
Reviewer 1 Report
The authors properly addressed my comments.
Reviewer 2 Report
I have further comments
Reviewer 3 Report
The authors responded promptly.